# Estetrol Combined to Progestogen for Menopause or Contraception Indication Is Neutral on Breast Cancer

**DOI:** 10.3390/cancers13102486

**Published:** 2021-05-20

**Authors:** Anne Gallez, Silvia Blacher, Erik Maquoi, Erika Konradowski, Marc Joiret, Irina Primac, Céline Gérard, Mélanie Taziaux, René Houtman, Liesbet Geris, Françoise Lenfant, Elisabetta Marangoni, Nor Eddine Sounni, Jean-Michel Foidart, Agnès Noël, Christel Péqueux

**Affiliations:** 1Laboratory of Biology, Tumors and Development, GIGA-Cancer, University of Liège, 4000 Liège, Belgium; anne.gallez@uliege.be (A.G.); silvia.blacher@uliege.be (S.B.); erik.maquoi@uliege.be (E.M.); e.konradowski@uliege.be (E.K.); irina.primac1@gmail.com (I.P.); nesounni@uliege.be (N.E.S.); jmfoidart@uliege.be (J.-M.F.); agnes.noel@uliege.be (A.N.); 2Biomechanics Research Unit, GIGA-In Silico Medicine, University of Liège, 4000 Liège, Belgium; marc.joiret@uliege.be (M.J.); liesbet.geris@uliege.be (L.G.); 3Mithra Pharmaceuticals, rue Saint-Georges 5/7, 4000 Liège, Belgium; cgerard@mithra.com (C.G.); mtaziaux@mithra.com (M.T.); 4Precision Medicine Lab, 5349 AB Oss, The Netherlands; rene@precisionmedicinelab.nl; 5INSERM U1048, Institut des Maladies Métaboliques et Cardiovasculaires, University Paul Sabatier, 31432 Toulouse, France; francoise.lenfant@inserm.fr; 6Translational Research Department, Institute Curie, PSL Research University, 75248 Paris, France; elisabetta.marangoni@curie.fr

**Keywords:** estetrol, estrogen receptor alpha, progesterone, drospirenone, breast cancer, menopause hormone therapy, combined oral contraceptive

## Abstract

**Simple Summary:**

Hormonal treatments, especially those used to treat menopause symptoms are known to increase breast cancer risk. It is thus necessary to identify new formulations with a better benefit/risk profile. The aim of this translational study was to evaluate the breast cancer risk associated with a combination of a natural estrogen named estetrol, with progestogens such as natural progesterone and drospirenone. Since the assessment of breast cancer risk in patients during drug development is not possible given the requirement of long-term studies in large populations, this study provides new evidence that a therapeutic dose of estetrol for menopause treatment or contraception, combined with progesterone or drospirenone, may provide a better benefit/risk profile toward breast cancer risk compared to the hormonal treatments currently available for patients.

**Abstract:**

Given the unequivocal benefits of menopause hormone therapies (MHT) and combined oral contraceptives (COC), there is a clinical need for new formulations devoid of any risk of breast cancer promotion. Accumulating data from preclinical and clinical studies support that estetrol (E4) is a promising natural estrogen for MHT and COC. Nevertheless, we report here that E4 remains active on the endometrium, even under a dose that is neutral on breast cancer growth and lung metastasis dissemination. This implies that a progestogen should be combined with E4 to protect the endometrium of non-hysterectomized women from hyperplasia and cancer. Through in vivo observations and transcriptomic analyses, our work provides evidence that combining a progestogen to E4 is neutral on breast cancer growth and dissemination, with very limited transcriptional impact. The assessment of breast cancer risk in patients during the development of new MHT or COC is not possible given the requirement of long-term studies in large populations. This translational preclinical research provides new evidence that a therapeutic dose of E4 for MHT or COC, combined with progesterone or drospirenone, may provide a better benefit/risk profile towards breast cancer risk compared to hormonal treatments currently available for patients.

## 1. Introduction

In western countries, it is estimated that 12 million women use menopause hormone therapy (MHT) [1]; moreover, 151 million women use a combined oral contraceptive (COC) worldwide [2]. The Collaborative Group on Hormonal Factors in Breast Cancer recently published a meta-analysis [1] revealing that: (i) half of the women with breast cancer have used MHT, (ii) an excess risk of breast cancer was associated with 1–4 years of use and progressively increased with MHT duration, (iii) the excess risk was greater for estrogen receptor (ER)-positive (ER+) than ER-negative (ER-) breast cancer, (iv) the risk was higher for estrogen-progestogen than for estrogen-only preparations. The progestogen is added to protect the endometrium against the proliferative effects of estrogens in non-hysterectomized menopausal women. The association of a possible increased breast cancer risk with the use of COCs is utterly difficult to determine. Nevertheless, several studies have reported that women using a COC have a slightly increased risk of developing breast cancer [3,4,5]. This slight risk disappears 10 years after treatment cessation, indicating that estrogens in these preparations promote preexisting breast cancer growth rather than induce breast carcinogenesis.

The fear of an increased risk of breast cancer due to the use of MHT or even a COC leads an increasing number of women to avoid these treatments, especially MHT. Given the unequivocal benefits of MHT and COCs to women’s health and well-being, there is a medical need for the development of new generation estrogen-progestogen preparations presenting a better safety profile, especially regarding the breast cancer risk. Estetrol (E4) could fulfill this medical need. This natural estrogen produced during pregnancy by the human fetal liver [6] has potential clinical applications in estrogen-sensitive tissues since it displays a binding specificity for ERs [7]. E4 administration in rodents and women revealed that E4 shares with estradiol (E2) and estriol (E3) several estrogen-like effects on numerous tissues. E4 reduces vasomotor symptoms [8], protects the endothelium from atheroma [9] and bones from osteoporosis [10]. It helps to avoid vaginal dryness as well [11]. E4 has also several characteristics distinctive from other estrogens, which makes it an appropriate compound to be used for MHT or COC in women. It has the longest half-life of naturally occurring estrogens (28–32 h), in contrast to E2 (t_1/2_ = 2–10 h) and E3 (t_1/2_ = 10–20 min) [12,13]. Contrary to ethinyl-estradiol (EE) and E2, E4 does not bind to sex-hormone-binding globulin (SHBG), and only moderately increases SHBG production in the human liver [14,15]. When combined with drospirenone (DRSP), the combination (15 mg E4/3 mg DRSP) showed reduced hemostatic effects compared to EE/DRSP combinations [16] and contraceptive efficacy in two large phase 3 trials that enrolled 2148 women in the USA and Canada and 1577 women in Europe and Russia [17,18]. A phase 1 study showed that 15 mg E4/day prevents hot flushes [8]. In addition, the capacity of E4 to abrogate the symptoms of menopause is currently being evaluated in two phase 3 studies.

E4 appears to be a weak estrogen on the mammary gland; when compared to E2 it induces only a moderate increase of proliferation even at high doses [19]. However, its full impact on breast tumorigenesis remains unknown, since it has been described to be pro-apoptotic but also pro-tumoral [20,21,22,23]. On one hand, E4 has been shown to decrease breast tumor growth in a 7,12-Dimethylben[a]antracene (DMBA)-induced breast cancer model in rats following a dose-dependent manner [20]. E4 was pro-apoptotic in vitro on long-term estrogen-deprived breast cancer cell lines [21,24]. In addition, when given for two weeks to women with recently diagnosed breast cancer, immunohistochemical assessment showed that 15 mg E4 increased the number of apoptotic cells when compared with placebo treatment. However, it did not change the Ki67 proliferation marker [22]. On the other hand, in preclinical breast cancer models in mice, E4 elicits dose-dependent estrogenic properties inducing pro-tumoral effects at high doses [23]. Interestingly, when E4 is combined with E2 it shows anti-estrogenic activity by decreasing the pro-tumoral effect of E2 [23,25].

The effects of the combination of E4 with a progestogen on breast malignancy have not been evaluated yet. In this study, using three complementary in vivo models of breast cancer (transgenic MMTV-PyMT mice, human breast adenocarcinoma MCF7 cell line xenografts, hormone-dependent breast tumor patient-derived xenografts (PDX)), we show that E4 combined with or without progesterone (P4) or DRSP promotes neither breast cancer development nor metastatic dissemination when used at a therapeutic dose for MHT or COC. The mechanistic insights we revealed by large scale coregulator recruitment assays, transcriptomic and protein analysis demonstrate a 50–100 times lower potency of E4 versus E2 to activate the recruitment of coregulators to ERα, to induce similar transcriptional activity and to sustain progesterone receptor (PR) expression in breast cancer. Moreover, the addition of progestogens shows only poor transcriptional impact. These key findings support that a therapeutic dose of E4 for MHT or COC, combined with or without P4 or DRSP, may provide a better benefit/risk ratio towards breast cancer risk compared to hormonal combinations currently available for patients.

## 2. Materials and Methods

### 2.1. Animals and Ethical Study Approval

Female swiss Nu/Nu mice were purchased from Charles River (Saint-Germain Nuelles, France). MMTV-PyMT (FVB/N strain) transgenic mice [26,27] were bred and maintained within the accredited Mouse Facility and Transgenics GIGA platform of the University of Liège (Liège, Belgium), under pathogen-free conditions. All animal experiments were conducted in accordance with the Federation of European Laboratory Animal Science Associations (FELASA) and were approved by the local ethical committee of the University of Liège.

### 2.2. Human Samples for PDX and Ethical Study Approval

Human hormone-dependent tumor samples from one patient were validated and provided by Prof. Marangoni (Institute Curie, Paris, France) for PDX experiments, as described previously [28,29]. The HBCx-131 PDX model was obtained by engrafting a biopsy from spinal bone metastasis of an ER+ breast cancer patient treated with vertebroplasty as detailed in Montaudon et al. [30]. The metastasis biopsy was engrafted with the patient’s informed consent into female Swiss nude mice (Charles River Laboratories, Saint-Germain Nuelles, France) that were maintained in specific pathogen-free animal housing and received estrogen diluted in drinking water. These experiments were conducted in accordance with the guidelines of the French Ethics Committee (project authorization no. 02163.02) and in accordance with the current legislation and recommendations of the Ethical Committee of the University Hospital of Liège (project authorization no. 14-1582).

### 2.3. Human Treatment Modeling in Mice: Dose Delineation

E2 was chosen as a reference treatment since it is recognized as the most widely used for MHT [1]. The steroid doses administered to mice were defined to match the human therapeutic dose of E2 and P4 recommend by the FDA and EMA for MHT, with E2 (0.5–2 mg/day) and P4 (100 or 200 mg/day) [31,32,33]. For E4 and DRSP the dose choice was based on clinical studies [17,18] (Appendix A).

### 2.4. Steroids and Reagents

17-beta estradiol (E2, #E8875), progesterone (P4, #P0130), drospirenone (DRSP, #SML0147) and propylene glycol were purchased from Sigma (Sigma Aldrich, Merck KGaA, Overrijse, Belgium). Estetrol (E4) was provided by Mithra Pharmaceuticals (Liège, Belgium). Ethanol (EtOH) was purchased from VWR Chemicals (VWR International, Leuven, Belgium).

### 2.5. Cell Cultures

Human breast cancer cells MCF7 (HTB-22™) and T47D (HTB-133™) were purchased from the American Type Culture Collection (ATCC, Manassas, VA 20110, USA). Both cell lines were authenticated by Leibniz-Institute DSMZ using STR DNA typing and Cytochrome Oxidase subunit 1 (COI) alignment, respectively. All cells were used within 10 passages after authentication. Cells were routinely cultured following the manufacturer’s instructions.

### 2.6. MMTV-PyMT, MCF7 Xenograft and PDX Mouse Models

All female MMTV-PyMT and Swiss Nu/Nu mice were bilaterally ovariectomized at four weeks of age (Figure 1A). For MMTV-PyMT, MCF7 xenograft mouse models, the following treatments were initiated two weeks after surgery: E2 administered through subcutaneous slow-releasing pellets (#ME2–60 days, Belma Technologies, Liège, Belgium) [34]; E4 (0.3 or 3 mg/kg/day, Appendix A) administered through subcutaneous Alzet^®^ pumps (model 2006, Charles River, Saint-Germain Nuelles, France); in some groups, E4 was combined with P4 (1.25 or 4.25 mg/kg/day) administered through subcutaneous slow-releasing pellets (#P4L-M/60 days or #P4-M/60 days, Belma Technologies, Liège, Belgium) or with DRSP (0.06 mg/kg/day) administered through subcutaneous Alzet^®^ pumps (Charles River, Saint-Germain Nuelles, France); untreated mice (OVX) were sham operated. One week later, MCF7 cells (4 × 10^5^ cells per flank, diluted 1:1 in Matrigel,) were subcutaneously injected in Swiss Nu/Nu mice.

For the PDX model, all Swiss Nu/Nu ovariectomized mice received a subcutaneous E2 pellet (#ME2–60 days, Belma Technologies, Liège, Belgium) two weeks after surgery, then human tumor fragments (2 mm^3^) were implanted in each flank one week after E2 treatment initiation. When tumors reached approximately 100 mm^3^, E2 treatment was removed and replaced by one of the above-described treatments used in the MMTV-PyMT and MCF7 xenograft models. Tumor size was assessed with a digital caliper and calculated as V (mm^3^) = length × width^2^ × 0.4.

### 2.7. Blood Sampling and Circulating E2, E4 and P4 Quantitation

Circulating levels of E4 in mice were measured as previously described [35]. Plasmatic concentrations of E2 and P4 were measured in the laboratory of Medical Chemistry and Clinical Study of the CHU of Liège.

### 2.8. Cell Proliferation

The proliferation of MCF7 and T47D cells was evaluated using the Cyquant™ Cell Proliferation Assay Kit (Life Technologies, Invitrogen, Thermo Fisher Scientific, Carlsbad, CA 92008, USA) after 24 h to 72 h treatment with E2, E4 or vehicle (EtOH 0.01%).

### 2.9. Lung Metastasis Quantification

To evaluate lung metastasis dissemination in MMTV-PyMT mice, paraffin-embedded lung sections (5µm) were stained with hematoxylin and eosin (H/E) and quantified by automated image analysis as described in the expanded Appendix A.

### 2.10. Protein Extraction and Western Blot Analysis

Proteins were extracted from MCF7 cells, MCF7 tumor xenografts and PDX tumors, then submitted to Western Blot using the following antibodies: anti-ERα (D8H8, #8644, Cell signaling Technology, Danvers, MA 01923, USA), anti-phospho S118-ERα (16J4, #2511, Cell signaling Technology, Danvers, MA 01923, USA), anti-PR (D8Q2J, #8757, C1A2, #3157, Cell signaling Technology, Danvers, MA 01923, USA), anti-GAPDH (#MAB374, Cell signaling Technology, Danvers, MA 01923, USA), anti-rabbit-HRP or anti-mouse-HRP (#7074, #7076, Cell signaling Technology, Danvers, MA 01923, USA).

### 2.11. RT-qPCR and RNAseq Analysis

Routinely cultured MCF7 and T47D cells were treated in red phenol-free DMEM medium (Gibco Invitrogen Corporation, UK), supplemented with 2% heat-inactivated and dextran-coated charcoal-treated FBS (FBS-cs, Lonza, Switzerland) with E2 (10^−9^M), E4 (10^−7^M or 10^−10^M) or vehicle (DMSO 0.001%) for 20 h, then P4 (10^−7^M) or R5020 (10^−8^M) were added in some conditions for four additional hours. After a 24 h total treatment, RNA was extracted using the High Pure RNA Isolation Kit (#11828665001, Roche Diagnostics GmbH, Mannheim, Germany) following the manufacturer’s protocol. RT-qPCR was performed with specific primers (Appendix A) and Roche Probes (Roche Diagnostics GmbH, Mannheim, Germany) on a LightCycler^®^ 480 Instrument and Software (Roche Diagnostics GmbH, Mannheim, Germany). For RNAseq analysis, total RNA extracted from five independent experiments was analyzed on an Illumina high throughput sequencer using the genomic platform in the GIGA organization of the University of Liège. RNAseq raw data are available on Gene Expression Omnibus (GEO; https://www.ncbi.nlm.nih.gov/geo/query/acc.cgi?acc=GSE173300, GEO accession: GSE173300, public on 17 May 2021). Data were analyzed with the Rstudio program version 1.1.463 (RStudio, Inc., Boston, MA 02210, USA). The analysis parameters used were: Fc ≥ 2, *p*-value ≤ 0.01 and power: 97%.

### 2.12. Immunohistochemical Staining for Ki67, ERα, pS118-ERα and PR

Uteri and tumors were collected at sacrifice and paraffin-embedded. Immunolabeling was carried out on serial 5 µm sections using anti-Ki67 (#Ab16667, Dako, Denmark), anti-ERα (SP1, #790-4325, Ventana, Roche Diagnostics GmbH, Mannheim, Germany), anti-phospho S118-ERα (16J4, #2511, Cell signaling Technology, Danvers, MA 01923, USA), anti-PGR (1E2, #790-4296, Ventana, Roche Diagnostics GmbH, Mannheim, Germany) antibodies, followed by the appropriate secondary anti-rabbit or anti-mouse Envision system -HRP antibodies (#K4003 and #K4001, Dako, Glostrup, Denmark). Image analysis quantifying tumor staining density was performed with Matlab software (MathWorks, Inc, Natick, MA 01760, USA) as previously described [36].

### 2.13. MARCoNI Assay

The ability of E2 and E4 to induce the binding of ERα to 154 co-regulator motifs was evaluated using the PamChip Microarray Assay for Real-Time Coregulator-Nuclear receptor Interaction (MARCoNI, PamGene International BV, HH’s-Hertogenbosch The Netherlands), as described previously. In this assay, arrays were incubated with crude protein extracts from MDA-MB-231 cells stably transfected with ERα [37] mixed with E2 or E4 at concentrations ranging from 10^−12^M to 10^−7^M. Binding of ERα to each co-regulator motif of the PamChip microarray was evidenced by Western Blot, as described previously [37]. MARCoNI raw data are available in Appendix A.

### 2.14. Statistical Analysis

Results were analyzed using GraphPad Prism 7.00 (GraphPad Software Inc., San Diego, CA 92108, USA). Unless otherwise stated, results were expressed as mean ± SEM. The equality of variance between groups was evaluated by the Shapiro–Wilk normality test and statistical tests were chosen accordingly. Student’s *t*-test or one-way analysis of variance (ANOVA) followed by Dunnett’s post-test were used for normal data distributions, otherwise, Mann–Whitney analysis, Kruskal–Wallis test followed by Dunns post-test or two-way ANOVA followed by Bonferroni post-test were used. The *p*-value was expressed as follows: * *p* < 0.05; ** *p* < 0.01; *** *p* < 0.001 and **** *p* < 0.0001.

### 2.15. Other Methods

Expanded methods are provided in Appendix A.

## 3. Results

### 3.1. Therapeutic Dose of E4 Stimulates Endometrial Proliferation

In order to model human treatments, it is essential to administer steroids to mice in a pattern that closely mimics steroid exposure in women (Appendix A). E2 has been chosen as a reference treatment since it is recognized as the most widely used for MHT [1]. E4 was used at two doses: (i) 0.3 mg/kg/day, which falls into the range of the plasma concentration of the E4 therapeutic dose (15 mg/day) for COC or MHT in women [8,17,18,38]; (ii) 3 mg/kg/day, corresponding to 10-fold the therapeutic dose (Appendix A). P4 (1.25 or 4.25 mg/kg/day) and DRSP (0.06 mg/kg/day) were used because they mimic the therapeutic doses used for women’s treatment [17,18,31,32]. For more details, see the dose delineation section of the materials and methods.

The uterotrophic activity of estrogens was used as a biological internal control of estrogen activity. Moreover, the efficacy of progestogen treatments to inhibit estrogen-mediated uterotrophic effects was controlled in all breast cancer models used to analyze the impact of estrogen-progestogen treatments on breast cancer growth. We used three complementary in vivo mice models of breast cancer: transgenic polyoma middle T oncogene-induced (MMTV-PyMT) mice (FVB/N genetic background), mice implanted with the human ER+ breast adenocarcinoma MCF7 cell line or with a hormone-dependent (ER+) breast PDX tumor (Figure 1A). All mice were ovariectomized (OVX) two weeks before starting treatments to mimic menopause and then treated with E4, in parallel to E2, combined or not with progestogens (P4 or DRSP). As expected, on MMTV-PyMT mice, E2 increased uterus wet weight (Figure 1B,C), luminal epithelial height (Figure 1B,D) and epithelial cell proliferation with 90% of cells staining positive for Ki67 (Figure 1B,E). A dose-dependent estrogenic effect was observed for E4 on epithelial cell proliferation, with 40% and 85% of cells positive staining for Ki67 after treatment with E4 0.3 mg/kg/day (therapeutic dose) and E4 3 mg/kg/day (supratherapeutic dose), respectively (Figure 1B,E). Both doses of P4 completely inhibited the proliferative effect of E2 and E4 on uterus epithelial cells (Figure 1B,E). DRSP (0.06 mg/kg/day) inhibited the proliferative effect of E4 (0.3 mg/kg/day) (Figure 1F–I). Similar results were obtained with the animals xenografted with MCF7 cells or PDX (Appendix A).

### 3.2. Therapeutic Dose of E4 Is Neutral on Breast Cancer Progression

First, we used the oncogene-driven model of transgenic MMTV-PyMT mice that recapitulates the different steps observed during the carcinogenesis of human luminal-like hormone-dependent breast cancer [26,27]. E2 treatment accelerated tumor appearance (Figure 2A), increased tumor growth starting after seven weeks of age (Figure 2B) and induced higher tumor mass (Figure 2C), when compared to untreated ovariectomized mice. In contrast to E2-treated mice, tumor appearance (Figure 2A), tumor growth (Figure 2B), tumor mass (Figure 2C) and tumor delay (Figure 2D) were not affected in mice receiving the therapeutic dose of E4 (0.3 mg/kg/day), when compared to untreated ovariectomized mice. Although the therapeutic dose of E4 (0.3 mg/kg/day) stimulated the proliferation of endometrial cells (Figure 1), it did not accelerate tumor progression. However, when used at the supratherapeutic dose (3 mg/kg/day), E4 induced effects similar to E2. Altogether, these results indicate that uterine epithelial cells are more sensitive to E4 than breast cancer cells.

The same treatments were applied to mice bearing human MCF7 cell xenografts. As expected, MCF7 tumors developed rapidly in E2-treated mice (Figure 2E,F). However, in contrast to E2 treatment, administration of the therapeutic dose of E4 (0.3 mg/kg/day) only allowed MCF7 engraftment but did not support tumor growth. The supratherapeutic dose of E4 (3 mg/kg/day) supported MCF7 tumor growth, albeit to a lesser extent when compared to E2. In a hormone-dependent PDX (Figure 2G,H), therapeutic E4 (0.3 mg/kg/day) did not increase PDX tumor growth, even after 30 weeks of treatment. By contrast, E2 or supratherapeutic E4 (3 mg/kg/day) rapidly promoted PDX growth (Figure 2G,H).

The impact of E4 treatment on lung metastasis dissemination from breast cancer cells was evaluated in the MMTV-PyMT model that spontaneously forms metastasis. When mice were treated with E2 or supratherapeutic E4 (3 mg/kg/day), 100% of them presented lung metastasis (Figure 2I,J). These treatments induced a mean number of 1.4 metastasis per cm^2^ of lung for both E2 and E4 (3 mg/kg/day) (Figure 2K), and mean sizes of 0.120 and 0.165 mm^2^ per metastasis for E4 (3 mg/kg/day) and E2, respectively (Figure 2L), corresponding to a total of 1–2% of the lung area (Figure 2M). In contrast, only 3/7 (43%) and 2/6 (33%) of mice presented with metastasis when treated with therapeutic E4 (0.3 mg/kg/day) or left untreated (OVX), respectively (Figure 2J). In addition, among the 43% of mice treated with therapeutic E4 (0.3 mg/kg/day) that had metastasis, the metastasis number was between 0.15 to 0.5 metastasis per cm^2^ of lung, their mean size was 10 times smaller (0.01 mm^2^) and they occupied less than 0.2% of the lung area (Figure 2I–M). Comparable results arose from untreated (OVX) or therapeutic E4 (0.3 mg/kg/day) treated animals; metastasis progression was suppressed in more than 50% or reduced by 10 times when compared to E2 treated animals.

Altogether, these results support the concept that a therapeutic dose of E4 does not affect hormone-dependent breast cancer progression, while it induces uterotrophic effects in the absence of a progestogen.

### 3.3. The Combination of a Therapeutic Dose of E4 with P4 Is Neutral on Breast Cancer Progression

To mimic MHT treatment of non-hysterectomized women, we combined each dose of E4 (0.3 or 3 mg/kg/day) with P4 (1.25 or 4.25 mg/kg/day) or with DRSP (0.06 mg/kg/day). These doses of progestogens were used because they mimic the therapeutic doses that are effective in humans [18,31,32]. E2 + P4 (P4 4.25 mg/kg/day) was used as a reference treatment for non-hysterectomized women.

In the three breast cancer models tested, the addition of P4 to E4 did not modify either tumor growth or tumor mass, when compared to E4 used alone (Figure 3A–L). Interestingly, treatment with therapeutic E4 (0.3 mg/kg/day) alone or combined with P4 (1.25 or 4.25 mg/kg/day) or with DRSP (0.06 mg/kg/day) did not affect tumor growth, in contrast to the large increase induced by E2 + P4 (Figure 3A,B,E,F,I,J). Moreover, the addition of P4 to supratherapeutic E4 (3 mg/kg/day) did not modulate tumor growth (Figure 3C,D,G,H,K,L). In MCF7 and PDX experiments, tumor growth in mice treated with a combination of supratherapeutic E4 (3 mg/kg/day) and P4 was consistently lower than with E2 + P4, the usual preparation for MHT (Figure 3G,H,K,L).

Furthermore, in contrast to E2 + P4, when mice were treated with therapeutic E4 (0.3 mg/kg/day) combined with P4 (1.25 mg/kg/day) or DRSP (0.06 mg/kg/day), only 33–38% of mice presented with metastasis (Figure 3M,N), similar to untreated mice (Figure 2J). Among the metastasis-positive mice treated with E4 0.3 mg/kg/day combined with P4 or DRSP, only 0.1–0.2% of the lung was invaded by metastasis (Figure 3O), which is 15 times lower than with E2 + P4 treatment. These results show that when compared to E2 + P4 treated animals, the addition of a therapeutic dose of P4 or DRSP to a therapeutic dose of E4 prevents metastasis formation in 60% of mice. In the remaining 40%, metastasis is reduced by 15 times. These results were similar to the ones obtained with untreated animals (OVX). However, the combination of supratherapeutic E4 (3 mg/kg/day) with P4 or DRSP induced effects similar to E2 + P4 on metastasis dissemination (Figure 3M,N) and on the surface of the lung invaded (Figure 3O). In addition, metastasis number (Figure 3P) and metastasis size (Figure 3Q) were similar to E2 + P4 and to the estrogen exposure alone (Figure 2K,L).

Altogether, these results further support the idea that the combination of a therapeutic dose of E4 with P4 or DRSP does not promote hormone-dependent breast cancer progression.

### 3.4. E4 Is Less Potent Than E2 in Promoting ERα Signaling in ER+ Breast Cancer

For a mechanistic understanding of the neutral effect of the therapeutic dose of E4 on breast tumor growth, we defined the potency of E4 to activate ERα signaling in vitro on MCF7 and T47D cells and in vivo in human MCF7 and PDX tumors. We evaluated the induction of progesterone receptor (*PGR*, gene; PR, protein) expression and the phosphorylation of ERα on serine 118 (S118), two well-known markers of estrogen-dependent ERα activation [39,40,41].

In vitro, E4 induced *PGR* mRNA expression in a dose-dependent manner starting at 10^−10^M or 10^−9^M for MCF7 and T47D cells, respectively. With 10^−7^M E4, *PGR* mRNA upregulation was similar to the one induced by E2 (10^−9^M) (Figure 4A,B). At a protein level, 10^−10^M E4 was not sufficient to stimulate PR production, but 10^−7^M E4 and 10^−9^M E2 similarly induced PRA and PRB, two isoforms of PR (Figure 4C,D). In addition, a reduction of the level of ERα expression was observed when cells were treated with 10^−9^M E2 or 10^−7^M E4, reflecting a negative feedback arising after the stimulation of the ERα pathway. In contrast, the level of ERα expression was maintained with 10^−10^M E4 treatment. Moreover, in contrast to E2 (10^−9^M), E4 used at 10^−10^M did not increase the proliferation of MCF7 and T47D cells (Figure 4E,F). However, higher concentrations of E4 (10^−9^ M to 10^−7^ M) exhibited a dose-dependent proliferative effect, whereby E4 only induced the same effect as E2 when used at a 100 times higher concentration than E2 (Figure 4G,H). These results support that, in contrast to 10^−9^M E2 and 10^−7^M E4, 10^−10^M E4 fails to activate ERα signaling in breast cancer cells.

To assess the E4 potency on ERα signaling in vivo, Western Blot and immunohistochemistry (IHC) were conducted on MCF7 and PDX tumors. ERα expression was maintained throughout the treatment period in MCF7 (Figure 5A,D) and PDX (Figure 5H,K) tumors. The expression of PR was lower in MCF7 tumors treated with therapeutic E4 (0.3 mg/kg/day) than with E2 or supratherapeutic E4 (3 mg/kg/day) (Figure 5A,B,D,E), suggesting a lower potency of therapeutic E4 in activating ERα signaling in vivo. Moreover, the phosphorylation of S118-ERα was also lower in MCF7 tumors treated with therapeutic E4 (0.3 mg/kg/day) than with E2 or supratherapeutic E4 (3 mg/kg/day) (Figure 5 A,C,D,F). Finally, the Ki67 index of proliferation was lower in MCF7 tumors treated with therapeutic E4 (0.3 mg/kg/day) than with E2 or supratherapeutic E4 (3 mg/kg/day) (Figure 5D,G). Furthermore, therapeutic E4 (0.3 mg/kg/day) was also less potent than E2 or supratherapeutic E4 (3 mg/kg/day) in inducing ERα signaling in PDX tumors. Indeed, PR expression and phosphorylation of S118-ERα were decreased in PDX tumors from mice treated with therapeutic E4, even after 30 weeks of treatment (Figure 5H–M), consistent with Ki67 index of proliferation (Figure 5K,N). The assessed markers for ERα signaling were similar in untreated (OVX) and in therapeutic E4-treated PDX tumors, although E2 and supratherapeutic E4 showed similar activation of ERα signaling (Figure 5H–N).

These results emphasize that therapeutic E4 displays a lower potency than E2 to induce ERα signaling in ER+ cancer cells in vivo. These observations explain why E4 is not potent enough to increase ER+ breast tumor growth at this dose.

### 3.5. The Therapeutic Dose of E4 Elicits No Transcriptional Activity, While a Supratherapeutic Dose of E4 Induces a Transcriptomic Profile Similar to E2 in Breast Cancer Cells

We delineated the transcriptomic profile induced in vitro by E4 and E2 on MCF7 cells by RNA sequencing (RNAseq) analysis of five biological replicates treated for 24h (Appendix A). To mimic the in vivo treatment conditions, MCF7 cells were treated with E2 (10^−9^M) and with two concentrations of E4: (i) 10^−10^M E4, mimicking the effect of the therapeutic dose in vivo, since it is the first concentration in vitro that did not increase breast cancer cell proliferation (Figure 4); (ii) 10^−7^M E4, corresponding to the supratherapeutic dose, since this concentration activated ERα signaling and increased breast cancer cell proliferation in vitro, similar to 10^−9^M E2 (Figure 4). Volcano plots comparing E2 and 10^−7^M E4 to vehicle conditions were similar (Figure 6A). The Ingenuity® Pathway Analysis (IPA) revealed that the two main biological functions associated with E2- or E4-gene signature were related to the positive regulation of the developmental process and to cell–cell signaling (Appendix A). However, cell migration and regulation of cell proliferation were predominantly associated with the E2-gene signature (Appendix A). Venn diagrams showed that compared to the vehicle, E2 and 10^−7^M E4 shared 80% of their transcriptional targets, with only a few genes being up- (37 versus 35) or downregulated (82 versus 69) by E2 or E4, respectively (Figure 6B). Among these genes (Appendix A), we evidenced the 10 main genes being mostly differentially up- or downregulated by E2 or E4, based on a threshold of adjusted *p*-value < 0.01 and Log_2_Fc > 1 (upregulated) or Log_2_Fc < −1 (downregulated) (Table 1). Moreover, the fold change of each gene was compared between E2 and 10^−7^M E4 treatments. A correlation of 96.74% highlighted that E2 and 10^−7^M E4 induced highly similar transcriptomic profiles with MCF7 cells (Figure 6C). A heatmap of differential gene changes in MCF7 cells highlighted that treatments with E2 and 10^−7^M E4 produced comparable gene expression profiles that differed considerably from the vehicle and 10^−10^M E4 (Figure 6D). In addition, the vehicle and 10^−10^M E4 exhibited highly similar gene expression profiles (Figure 6D). These observations were confirmed by a volcano plot analysis comparing either E2 versus 10^−7^M E4 or 10^−10^M E4 versus vehicle, where no difference in gene expression was observed when comparing these conditions (Figure 6E).

Altogether, these results support that 10^−10^M E4 (therapeutic) does not induce any transcription activity in breast cancer cells, corroborating the lack of ERα signaling upregulation observed by Western Blot and IHC staining in vivo. However, 10^−7^M E4 (supratherapeutic) induced a transcriptomic profile being 97% similar to the E2-dependent one.

### 3.6. The Addition of P4 to E4 Induces Poor Transcriptional Activity in Breast Cancer Cells

To further characterize the impact of the addition of a progestogen to E4, we defined by RNAseq the transcriptomic profile induced by the addition of P4 to the therapeutic (10^−10^M) or the supratherapeutic (10^−7^M) dose of E4 on MCF7 cells. Moreover, we also tested R5020, a synthetic and more stable analog of P4 (Appendix A). Venn diagrams showed that 79% of the transcriptional targets induced by P4 were similar to the ones induced by R5020 irrespective of the estrogen used (10^−7^M E4 or E2, Figure 6F,G). However, the expression of only a few genes was modulated by combinations of 10^−10^M E4 with P4 or R5020, confirming the low signaling potency of 10^−10^M E4 (Figure 6H). Comparing the addition of P4 to E2 or 10^−7^M E4, we observed an overlap of 438 regulated genes, corresponding to 75–79% of common transcriptional targets (Figure 6I). Similar results were obtained by adding R5020 to E2 or 10^−7^M E4 (Figure 6J). However, when P4 or R5020 were added with 10^−10^M E4, only four and one genes were upregulated, respectively (Figure 6K,L). Nevertheless, compared to E2 or E4 alone, the addition of P4 or R5020 to E2 or 10^−7^M E4 led to the upregulation of a maximum of 17 genes, among which there was a majority of common genes (Figure 6M). Otherwise, their addition to 10^−10^M E4 upregulated only two genes (Figure 6M).

Altogether, these results emphasize that the addition of P4 to E4 has a very limited transcriptional impact on breast cancer cells, corroborating the neutral effect observed after the addition of a progestogen to E4 in vivo.

### 3.7. E4 Is Less Potent Than E2 in Inducing the Recruitment of Co-Regulators to ERα in Breast Cancer Cells

Coregulators are critical determinants of ERα-mediated transcriptional regulation [42]. Therefore, to further characterize the impact of E4 on ERα signaling, we compared the efficacy and the potency of increasing concentrations of E4 or E2 (10^−12^M to 10^−5^M) to induce the recruitment of coregulators to ERα. We used the cell-free MARCoNI (Microarray Assay for Real-time Co-regulator Nuclear receptor Interaction) assay system [43,44] allowing the characterization of the interaction of ERα with 154 different binding motifs derived from 64 different nuclear receptor coregulators. The heatmap showing hierarchical clustering and comparing the recruitment pattern induced by E4 and E2 revealed that both estrogens induced the binding between ERα and a similar subset of coregulators (Figure 7A, Appendix A). As expected, E2 induced the recruitment of well-characterized ERα coregulators [42,45] such as the co-activators mediator complex subunit 1 (MED1), proline-, glutamic acid- and leucine-rich protein 1 (PELP1), steroid receptor coactivator (SRC) 1, SRC2, SRC3, cAMP response element-binding protein (CBP/p300), bromodomain-containing protein 8 (BRD8) or the co-repressors ligand-dependent corepressor (LCoR) and nuclear receptor-interacting protein 1 (NRIP1/RIP140) (Figure 7B). E4 also induced the recruitment of these coregulators to ERα, although the potency of E4 was lower than that of E2. In addition, the slope of these binding curves was very sharp when binding was induced by E2, although it was softened with E4. The mean EC50 values were 10^−8^M for E4 and 0.5 × 10^−10^M for E2 (Figure 7C). These results show that, compared to E2, E4 induces recruitment by ERα of a similar subset of co-regulators, although with a lower potency. Nevertheless, the slope of these binding curves indicates that the ability of E4 to activate ERα signaling is more progressive than the one of E2, allowing a larger window of dose adjustment.

## 4. Discussion

The increase in breast cancer risk in estrogen-progestogen MHT users is the result of a growth acceleration of existing breast cancer cells [1]. However, the assessment of this risk during the development of new MHT generations is not possible given the requirement of extensive and long-term studies in large populations. To overcome this clinical issue, it is important to evaluate the potential impact of new MHTs using robust animal models of breast cancer. E4 is a natural fetal estrogen with specific features supporting an increased beneficial/risk profile in comparison to E2 or EE. It is currently being developed for contraception and menopause indications, but its effect on breast cancer when combined with a progestogen has not previously been assessed. Addressing this issue is mandatory before the transition to the clinic since the majority of women taking hormonal treatment are not hysterectomized and are treated with estrogen-progestogen formulations.

In the present study, we first compare the impact of E4 therapeutic and supratherapeutic doses on breast cancer development and progression. Specific attention was paid to modeling hormonal human treatments by administering steroids to mice in a pattern that closely mimics steroid exposure in women. We demonstrate that E4, administered continuously to mimic the steady-state plasma concentrations observed in women, affects neither breast cancer growth nor metastatic dissemination to the lung when used at 0.3 mg/kg/day. This dose is within the range of E4 levels circulating in the blood when administered orally at 2–20 mg/day, corresponding to the dose used in clinical trials [38]. Treatment of post-menopausal women with 15 mg E4/day reduces hot flushes [8] and the combination of 15 mg E4/3 mg DRSP has shown contraceptive efficacy [17,18]. Our results support that when E4 is administered at a therapeutic dose for menopause or contraception, it is neutral on breast cancer growth. Nevertheless, when E4 is used at 10-fold the therapeutic dose (3 mg/kg/day), it exerts pro-tumoral activity similar to that observed with E2. Similarly, when E4 was administered by oral gavage [23], a dose of 10 mg/kg/day was necessary to achieve a pro-tumoral effect identical to that of 3 mg/day E2, although 0.5 mg/kg/day E4 remained neutral.

In contrast to the dose-dependent anti-tumoral effect of E4 described in a DMBA-induced breast cancer rat model [20], we did not observe any anti-tumoral activity of high or low dose E4 in the three breast cancer models evaluated. However, we used ovariectomized mice to mimic menopause, while the rats were not ovariectomized and had endogenous levels of E2 [20,46]. The anti-tumoral activity reported on the rat model could be ascribed to the anti-estrogenic properties of E4 that were also reported when E4 was combined with E2 (i) in MCF7 xenografts [23], (ii) on migration and invasion of T47D breast cancer cells [47] and (iii) in some ER+ breast cancer patients [22]. Altogether, these results are promising for the development of a new MHT or COC based on E4 with limited to no impact on breast cancer progression.

Our results highlight that ER+ breast cancer cells are less sensitive to E4 than uterine epithelial cells since the therapeutic E4 dose was sufficient to increase the proliferation of uterine epithelial cells. The neutral effect of the therapeutic dose of E4 on ER+ breast cancer growth is associated with a lack of ERα signaling activation in breast cancer cells. Especially, we show that the induction of PR expression and S118-ERα phosphorylation, two markers of ERα signaling activation, requires a supratherapeutic dose of E4 to achieve the same effect as the therapeutic dose of E2 in breast cancer cells in vitro and in vivo. Through a large-scale transcriptomic analysis comparing E4 and E2, we have demonstrated that 10^−10^M E4 is not sufficient to induce any transcriptional activity in MCF7 cells. Our results suggest that the pro-apoptotic properties of E4 evidenced in the phase 1 human study by Singer et al. [22] and in MCF7 cells treated with 10^−12^M E4 [21] relies on a transcriptionally-independent activity.

E2 and E4 induce 97% of common genes when the E4 concentration is 100 times higher than the E2 one, 10^−7^M and 10^−9^M, respectively. Interestingly, among the biological pathways associated with both gene signatures, cell migration and regulation of cell proliferation were predominantly associated with the E2-gene signature. In addition, the 3% of genes specifically regulated by E4 were not particularly related to any pro-tumoral functions or pathways, suggesting there is no potential increase of adverse pro-tumoral effect of E4 in comparison to E2. Interestingly, E4 predominantly upregulated ALX4, a tumor suppressor transcription factor downregulated in breast cancer cell lines such as MCF7 and in 70% of breast cancer from patients [48]. Finally, by evaluating the interaction between ERα and 154 coregulator motifs, which are key determinants controlling ERα-mediated transcriptional regulation [42], we found that the binding of either E2 or E4 to ERα recruits a similar subset of coregulators when the E4 concentration is 50 times higher than the E2 one. Nevertheless, the slope of these binding curves indicates that E2 only needs a slight increase of concentration to reach its maximal activity, although E4-related ERα activation is more progressive. Altogether, these results are in line with E4 being a weak estrogen that only induces ERα signaling at supratherapeutic doses in ER+ breast cancer, providing a larger window of therapeutic opportunity than with E2. The activation by E2 or E4 of ERα signaling contributes to enhancing breast cancer proliferation. It is therefore plausible that the absence of membrane ERα activation by E4, as documented by Abot et al. [9] in the endothelium, contributes to the differential effect of E4 versus E2 on breast cancer cells in vitro and tumor progression in vivo.

We observed that E4 remains active on the endometrium, by stimulating the proliferation of endometrial epithelial cells, even under a chronic therapeutic dose that is neutral on breast cancer growth and lung metastasis dissemination. This observation has a major consequence in the clinic since it implies that a progestogen should be combined with E4 to protect the endometrium of non-hysterectomized women from hyperplasia and cancer.

A key finding in the present study is that the addition of P4 or DRSP to E4 remains neutral on the three complementary in vivo ER+ breast cancer models we tested. These preclinical observations are particularly interesting in light of the meta-analysis of clinical data showing a higher excess risk reported for currently used estrogen-progestogen than for estrogen-only preparations [1]. Our results suggest that combining a progestogen with E4 could be safer than when it is combined with E2 or EE. In a preclinical study, P4 and a synthetic progestin R5020 were shown to decrease the E2-stimulated proliferation of breast cancer [49]. However, we did not observe any anti-proliferative effects of P4 or DRSP in any of the E4-treated ER+ breast tumors we evaluated. This could be related to the differences in steroid doses used in both studies or to the lower potency of E4 in comparison to E2 to induce ER/PR crosstalk. Nevertheless, we report that some genes (SGK1, FAM105A, FGF18, TMEM63C) were included in the gene signature as evidenced by Mohammed et al. [49] and associated with the P4-induced anti-proliferative effect, are also modulated in treatments combining E4 and P4 or R5020. These observations support that the estrogen/progestogen dose ratios are worth studying with more attention in breast cancer. Altogether, those results prompt us to consider treatments combining E4 with P4 or DRSP as a safer alternative MHT for non-hysterectomized women.

The main limitations of this study rely on the use of experimental animal models with restrictions in treatment duration. To largely cover the different stages of breast cancer development, we used the MMTV-PyMT mice [26,27] that develop breast tumors recapitulating the different steps observed during the carcinogenesis of human luminal-like hormone-dependent breast cancer and that allow the assessment of metastasis dissemination to the lung. In addition, we used the human ER+ adenocarcinoma MCF7 cell line xenograft and hormone-dependent PDX. PDX is a powerful tool for understanding breast cancer characteristics and for predicting drug potency [46,50]. Even if PDX does not fully recapitulate all the aspects of the human disease, especially the immune contribution, PDX maintains the original features of patient tumors and reflects drug sensitivity. It allows the treatment of human breast tumors in in vivo conditions since the tumor is submitted to blood circulating levels of the drug tested. Nevertheless, hormone-dependent PDX remains rare and difficult to generate compared to triple negative PDX [28]. To mimic as close as possible MHT treatments that last several years for women, treatments that did not increase PDX growth were maintained for 30 weeks. This corresponds to ¼ of a mouse’s life.

## 5. Conclusions

The clinical development of MHT and COC based on E4 is ongoing. Phase 3 clinical studies were completed for COC and are ongoing for MHT. Nevertheless, the assessment of these treatments on breast cancer risk in women can only be conducted during patient follow-up over decades. In this preclinical study, we show that E4 is neutral on breast cancer development when administered at the therapeutic dose for MHT or COC, even when it is combined with P4 or DRSP. Therefore, these results emphasize that the therapeutic dose of E4 combined with or without P4 or DRSP presents a better benefit/risk profile, especially towards breast cancer risk.

## Figures and Tables

**Figure 1 cancers-13-02486-f001:**
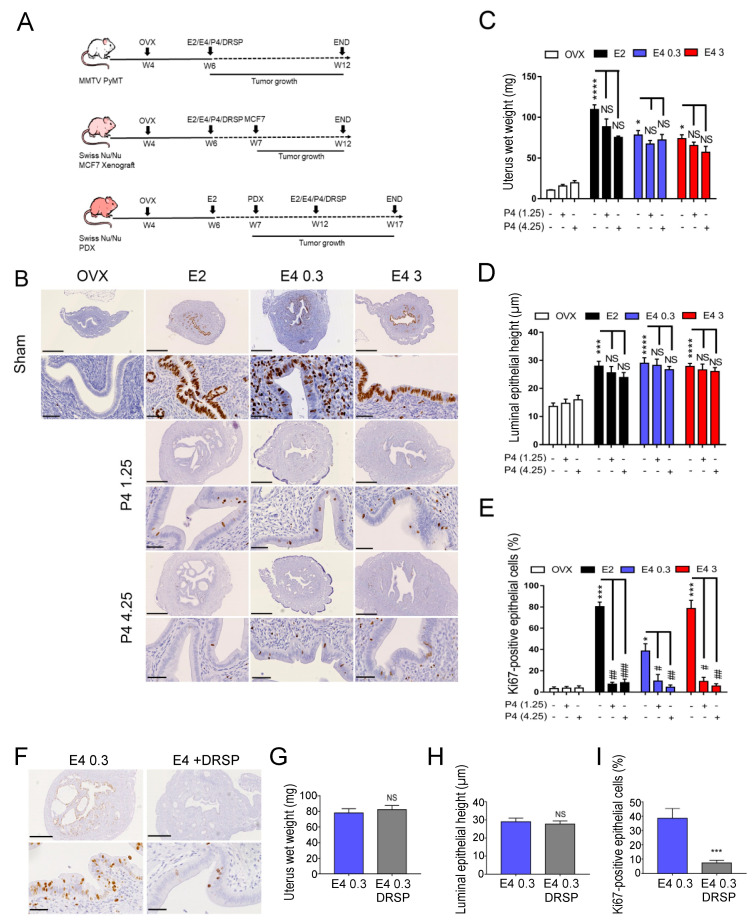
Uterotrophic effect of estetrol (E4), estradiol (E2), progesterone (P4) and drospirenone (DRSP). **(A**) Treatment protocol schema of the three hormone-dependent breast cancer mouse models: MMTV-PyMT, MCF7 xenograft and patient-derived xenograft (PDX). OVX, ovariectomy; E2/E4/P4/DRSP treatment start pointed by arrows; MCF7, tumor cell injection; PDX, tumor graft; END, mouse sacrifice; W4-W17, 4–17 weeks of age. (**B**) Representative Ki67 immunostainings on uterus harvested from MMTV-PyMT mice untreated (OVX) or treated with E2, E4 (0.3 or 3 mg/kg/day) combined with or without P4 (1.25 or 4.25 mg/kg/day); scale bar = 500 µm, zoom scale bar = 50 µm. Quantification of (**C**) uterine wet weight, (**D**) luminal epithelial height and (**E**) epithelial cell proliferation (Ki67-positive staining). (**F**) Representative Ki67 immunostainings on uterus harvested from MMTV-PyMT mice treated with E4 (0.3 mg/kg/day) with or without DRSP (0.06 mg/kg/day); scale bar = 500 µm, zoom scale bar = 50 µm. Quantification of (**G**) uterine wet weight, (**H**) luminal epithelial height and (**I**) epithelial cell proliferation (Ki67-positive staining). Kruskal–Wallis analysis followed by Dunn’s post-tests or Mann–Whitney analysis, *n* = 6–8 mice/condition. NS: not statistically significant; * or #: *p* < 0.05; ** or ##: *p* < 0.01; *** or ###: *p* < 0.001 and **** or ####: *p* < 0.0001. * versus OVX, # or NS versu*s* corresponding sham/estrogen-alone treated mice.

**Figure 2 cancers-13-02486-f002:**
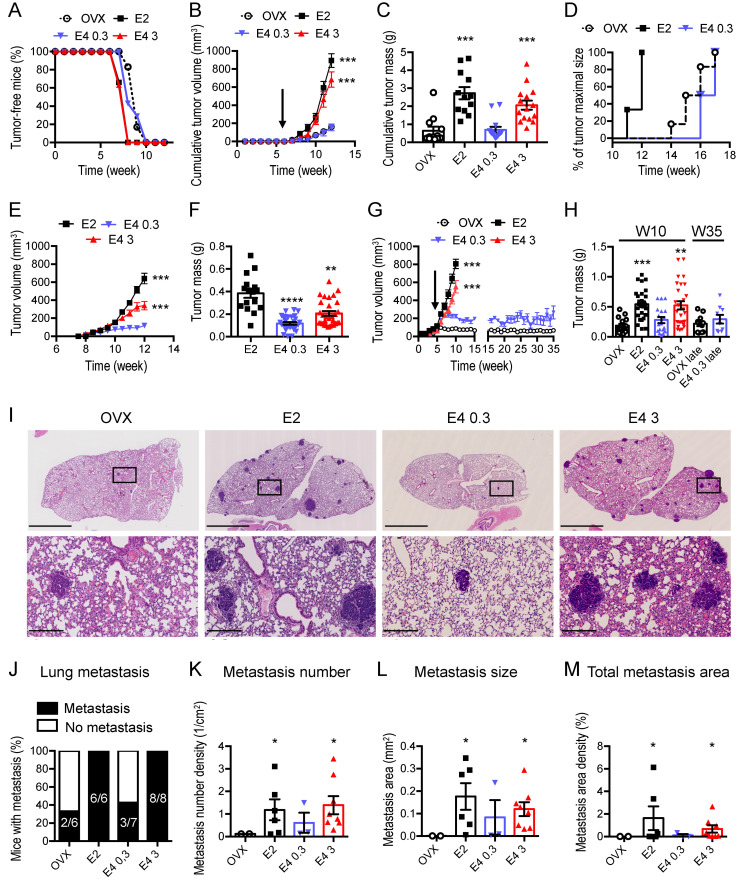
Dose-dependent effect of E4 on breast cancer progression**.** (**A**) Tumor appearance assessed by the percentage of tumor-free mice in ovariectomized MMTV-PyMT treated with E2 (0.08 mg/kg/day) or E4 (0.3 or 3 mg/kg/day). (**B**) Tumor growth kinetics (treatment start pointed by the arrow), (**C**) tumor mass and (**D**) tumor growth delay. (**E**) Tumor growth kinetics and (**F**) tumor mass of MCF7 xenografts from mice untreated (OVX) or treated with E2 (0.08 mg/kg/day) or E4 (0.3 or 3 mg/kg/day). (**G**) Tumor growth kinetics (treatments started five weeks after engraftment as pointed by the arrow) and (**H**) tumor mass of PDX from mice untreated (OVX) or treated with E2 (0.08 mg/kg/day) or E4 (0.3 or 3 mg/kg/day) for 5 (W10) or 30 weeks (W35). (**I**) Hematoxylin/eosin staining of lungs harvested from MMTV-PyMT mice; scale bar = 2.5 mm, zoom scale bar = 250 µm. (**J**) Percentage of metastasis-positive mice at sacrifice, (**K**) metastasis number, (**L**) metastasis size, (**M**) lung area occupied by metastasis (%). Kruskal–Wallis analysis followed by Dunn’s post-tests, two-way ANOVA analysis followed by Tukey post-tests or Mann–Whitney analysis, *n* = 5–15 mice/condition. *: *p* < 0.05; **: *p* < 0.01; ***: *p* < 0.001; ****: *p* < 0.0001. * versus OVX.

**Figure 3 cancers-13-02486-f003:**
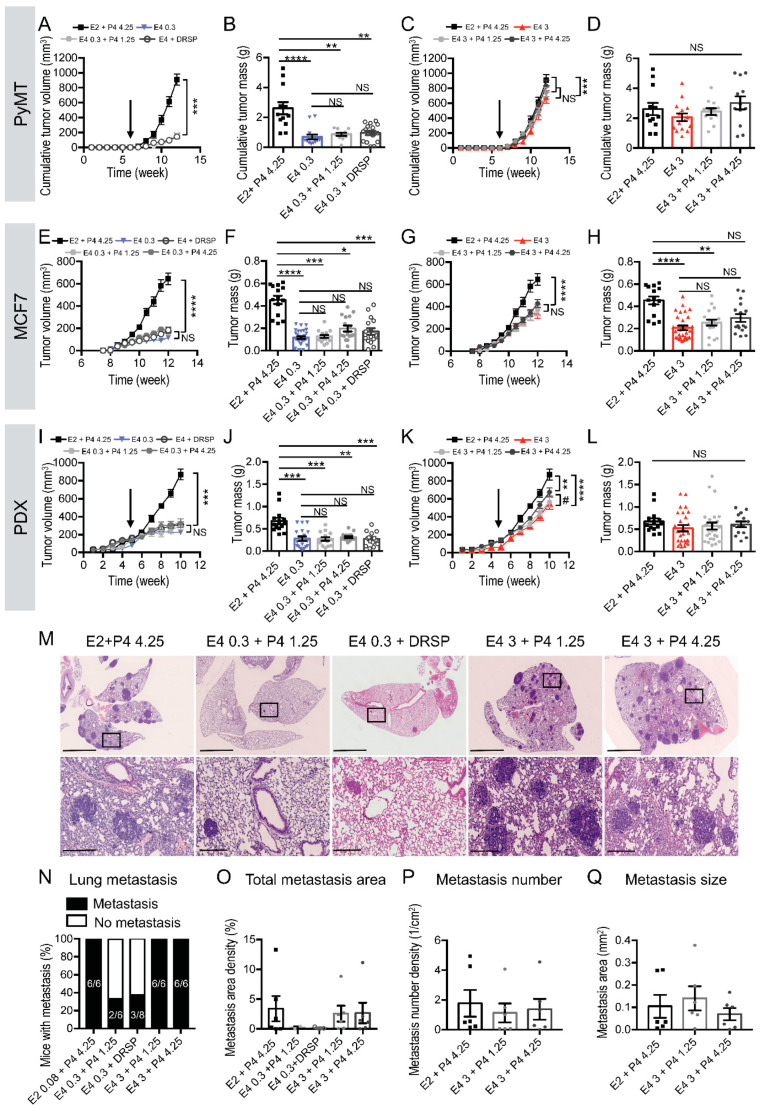
Dose-dependent effect of E4+progestogen on breast cancer progression**.** (**A**) Tumor growth kinetics (treatment start pointed by the arrow) and (**B**) tumor mass of MMTV-PyMT mice treated with E2 + P4 (4.25 mg/kg/day), E4 (0.3 mg/kg/day) combined with or without P4 (1.25 mg/kg/day) or DRSP (0.06 mg/kg/day). (**C**) Tumor growth kinetics (treatment start pointed by the arrow) and (**D**) tumor mass of MMTV-PyMT mice treated with E2 + P4 (4.25 mg/kg/day), E4 (3 mg/kg/day) combined with or without P4 (1.25 or 4.25 mg/kg/day). (**E**) Tumor growth kinetics and (**F**) tumor mass of MCF7 xenografts from mice treated with E2 + P4 (4.25 mg/kg/day), E4 (0.3 mg/kg/day) combined with or without P4 (1.25 or 4.25 mg/kg/day) or DRSP (0.06 mg/kg/day). (**G**) Tumor growth kinetics and (**H**) tumor mass of MCF7 xenografts from mice treated with E2 + P4 (4.25 mg/kg/day), E4 (3 mg/kg/day) combined with or without P4 (1.25 or 4.25 mg/kg/day). (**I**) Tumor growth kinetics (treatments started five weeks after engraftment as shown by the arrow) and (**J**) tumor mass of PDX from mice treated with E2 + P4 (4.25 mg/kg/day), E4 (0.3 mg/kg/day) combined with or without P4 (1.25 or 4.25 mg/kg/day) or DRSP (0.06 mg/kg/day). (**K**) Tumor growth kinetics (treatment start pointed by the arrow) and (**L**) tumor mass of PDX from mice treated with E2 + P4 (4.25 mg/kg/day), E4 (3 mg/kg/day) combined with or without P4 (1.25 or 4.25 mg/kg/day). (**M**) Hematoxylin/eosin coloration of lungs harvested from MMTV-PyMT mice; scale bar = 2.5 mm, zoom scale bar = 250 µm. (**N**) Percentage of metastasis-positive mice at sacrifice, (**O**) lung area occupied by metastasis (%), (**P**) metastasis number, (**Q**) metastasis size. Kruskal-Wallis analysis followed by Dunn’s post-test, two-way ANOVA analysis followed by Tukey post-tests or Mann Whitney analysis, *n* = 6–13 mice/condition. *: *p* < 0.05; **: *p* < 0.01; ***: *p* < 0.001; ****: *p* < 0.0001, * versus E2 + P4.

**Figure 4 cancers-13-02486-f004:**
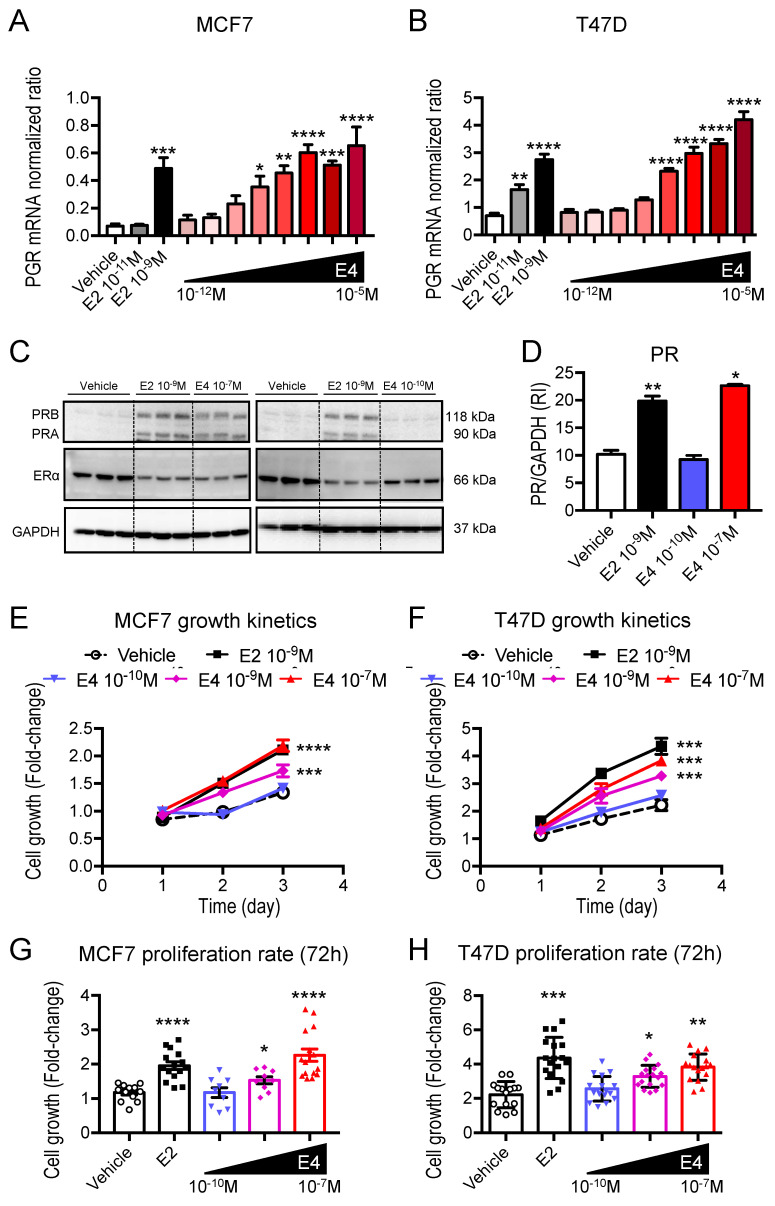
E4-induced ERα signaling in vitro**.**
*PGR* mRNA expression normalized to *TBP* and *GAPDH* in MCF7 (**A**) and T47D cells (**B**) treated with vehicle (EtOH 0.01%), E2 (10^−11^M or 10^−9^M) or E4 (ranging from 10^−12^M to 10^−5^M) for 4 h. *n* = 3–4 independent experiments. One-way ANOVA analysis followed by Dunnett’s post-test. *: *p* < 0.05; **: *p* < 0.01; ***: *p* < 0.001; ****: *p* < 0.0001, * versus vehicle. (**C**) Representative Western Blot of PR (PRA and PRB) and ERα from MCF7 cells treated with vehicle (EtOH 0.01%), E2 (10^−9^M) or E4 (10^−7^M or 10^−10^M) for 24h. GAPDH was used as a loading control. (**D**) Quantification of PR expression normalized to GAPDH level, RI= Relative Intensity, *n* = 3 independent replicates. (**E, F**) Representative experiment of cell growth kinetics of MCF7 and T47D cells treated with vehicle (EtOH 0.01%), E2 (10^−9^M) or E4 (10^−10^M, 10^−9^M or 10^−7^M). (**G, H**) Proliferation rate after 72 h. Mann–Whitney test. *: *p* < 0.05; **: *p* < 0.01; ***: *p* < 0.001; ****: *p* < 0.0001, * versus vehicle.

**Figure 5 cancers-13-02486-f005:**
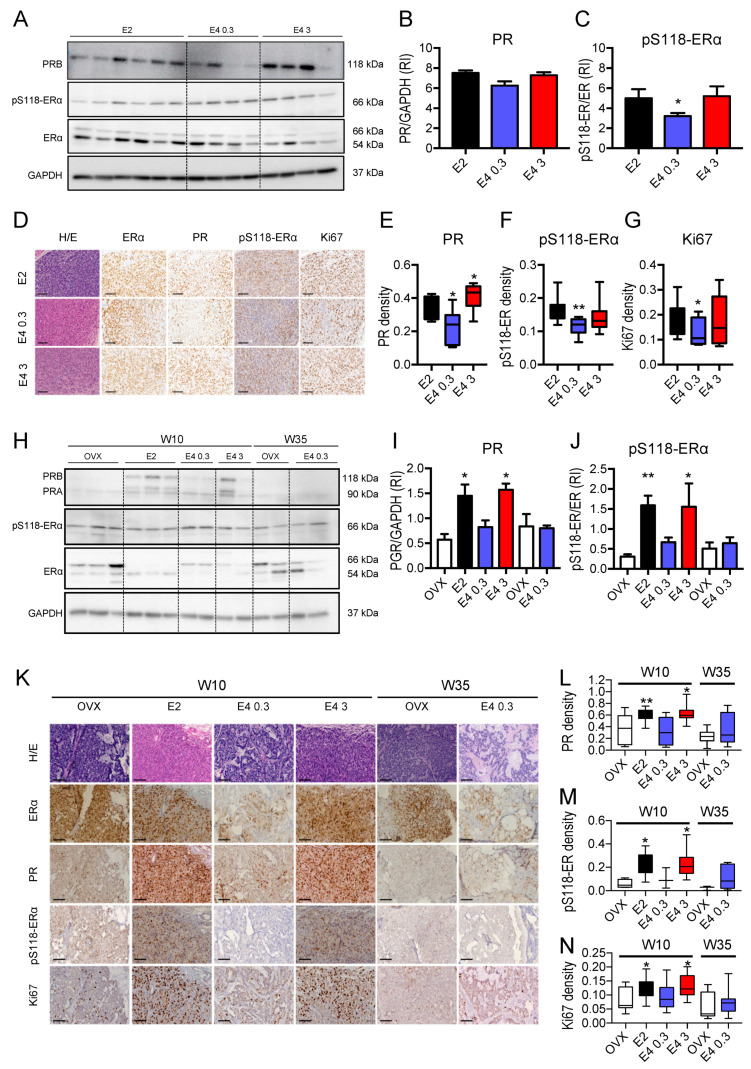
E4 is less potent than E2 to promote ERα signaling in vivo. (**A**) Representative Western Blot of PRB, pS118-ERα and ERα from MCF7 tumors harvested from mice treated with E2 (0.08 mg/kg/day) or E4 (0.3 or 3 mg/kg/day). GAPDH was used as a loading control. (**B**) Quantification of PR expression normalized to GAPDH level and (**C**) quantification of pS118-ERα protein level normalized to ERα. RI= Relative Intensity. (**D**) Representative immunostainings of ERα, PR, pS118-ERα and Ki67 on MCF7 tumors harvested from mice treated with E2 (0.08 mg/kg/day) or E4 (0.3 or 3 mg/kg/day); scale bar = 100 µm. Quantification of (**E**) PR, (**F**) pS118-ERα and (**G**) Ki67 staining expressed as density by Minimum and Maximum boxes. (**H**) Western Blot of PR (PRA and PRB), pS118-ERα and ERα from PDX untreated (OVX) or treated with E2 (0.08 mg/kg/day) or E4 (0.3 or 3 mg/kg/day) for 5 (W10) or 30 weeks (W35). GAPDH was used as a loading control. Quantification of (**I**) PR synthesis normalized to GAPDH level and quantification of (**J**) pS118-ERα normalized to ERα. (**K**) Representative immunostainings of ERα, PR, pS118-ERα and Ki67 from PDX untreated (OVX) or treated with E2 (0.08 mg/kg/day) or E4 (0.3 or 3 mg/kg/day) for 5 (W10) or 30 weeks (W35), scale bar = 500 µm. Quantification of (**L**) PR, (**M**) pS118-ERα and (**N**) Ki67 staining expressed as density by Minimum and Maximum boxes. Mann–Whitney tests, *n* = 8–12 tumors/condition. *: *p* < 0.05; **: *p* < 0.01, * versus E2 in MCF7, * versus OVX in PDX.

**Figure 6 cancers-13-02486-f006:**
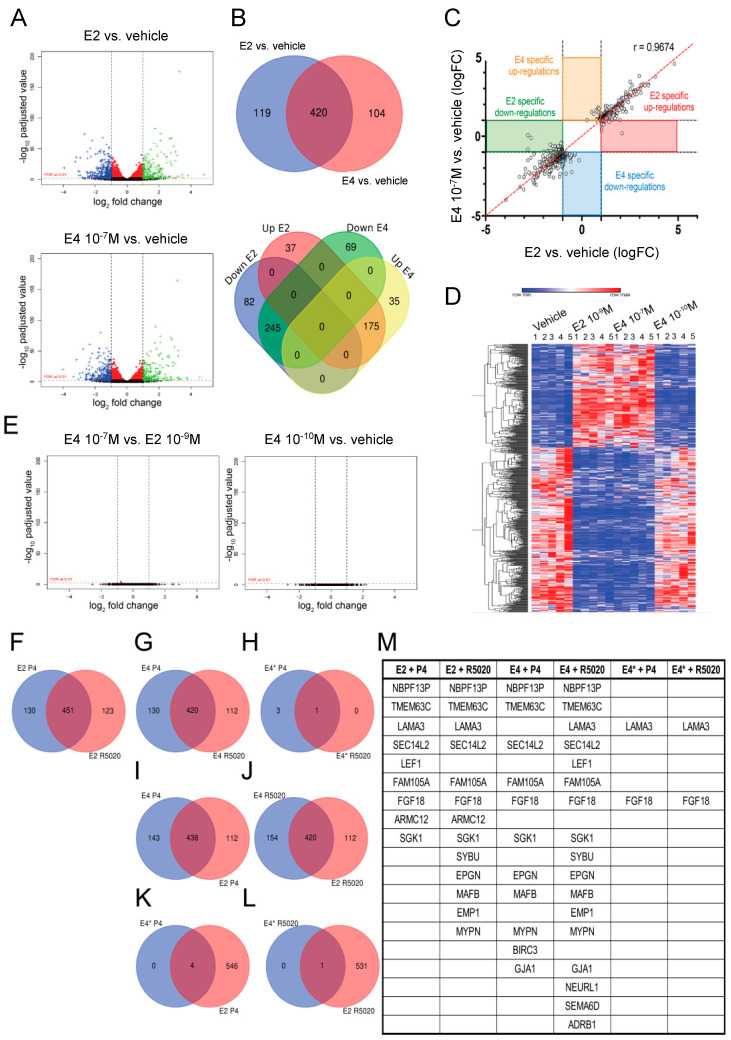
E2 and E4 transcriptomic profiles. (**A**) Volcano plot, using a parametric edgeR approach to identify differentially expressed (DE) genes, comparing E2 or E4 (10^−7^M) to vehicle from five independent replicates per condition. (**B**) Venn diagram comparing up- and downregulated genes by E2 or E4 (10^−7^M). (**C**) Correlation between genes regulated by E2 and/or E4 (10^−7^M) treatments. (**D**) Heatmap of gene regulation for each replicate of the different treatments: vehicle, E2 (10^−9^M) and E4 (10^−10^M, 10^−7^M). (**E**) Volcano plot, using a parametric edgeR approach to identify DE genes, comparing E4 (10^−7^M) to E2 and E4 (10^−10^M) to vehicle. (**F–L**) Venn Diagrams comparing genes regulated by E2 + P4, E2 + R5020, E4 + P4, E4 + R5020, E4* + P4, E4* + R5020 versus vehicle, E4 = 10^−7^M E4, E4* = 10^−10^M E4. (**M**) Specific genes modulated by E2 + P4, E2 + R5020, E4 + P4, E4 + R5020, E4* + P4, E4* + R5020 in comparison with respective estrogenic treatment alone. The analysis parameters used were: Fc ≥ 2, *p*-value ≤ 0.01 and power: 97%.

**Figure 7 cancers-13-02486-f007:**
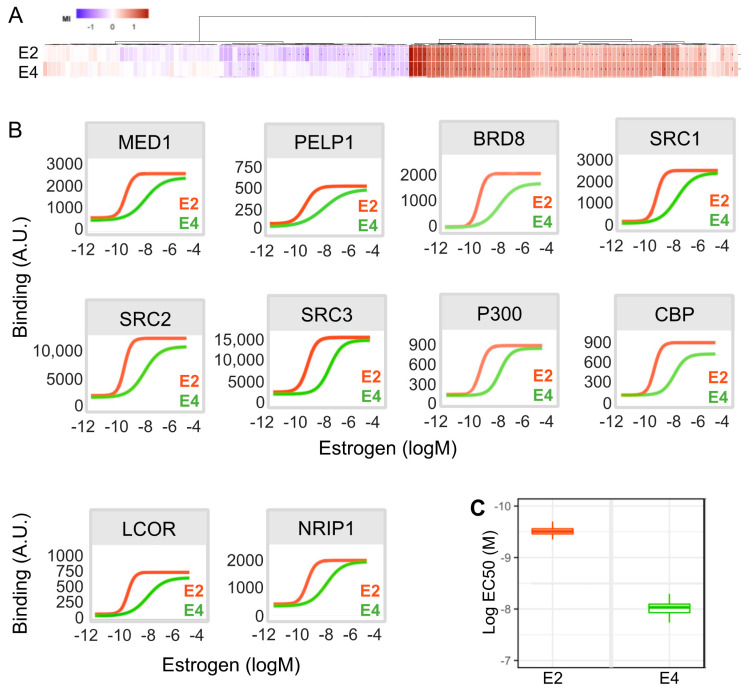
E2- and E4-specific co-regulator binding profiles. (**A**) Heatmap of interactions between ERα and co-regulators induced by E2 or E4 and represented as the modulation index (MI). MI is expressed as a log of fold-changes relative to the vehicle. (**B**) Dose-dependent induction by E2 or E4 (10^−12^M to 10^−5^M) of the ERα interaction with 10 co-regulators. (**C**) Boxplot comparing the mean of all EC50 (logM) values obtained with E2 or E4.

**Table 1 cancers-13-02486-t001:** E2- and E4-gene signatures.

	E2			E4		
	Gene	Log_2_ Fc	Adj *p*-value	Gene	Log_2_ Fc	Adj *p*-value
Upregulation	LINC01426	2.60402	0.00148	ALX4	3.42629	0.00009
	PCDHB7	2.31027	0.00546	IGF2	2.46035	0.00923
	CTB-178M22.2	2.08914	0.00193	DCLK2	1.91340	0.00299
	OTOF	2,33653	0.00589	AARD	1.89141	0.00188
	KLRG1	1.86945	0.00994	EREG	1.87651	0.00002
	MIR3153	1.81026	0.00603	F5	1.81722	0.00770
	KCNRG	1.72432	0.00064	CCDC73	1.81317	0.00605
	CFAP58-AS1	1.67121	0.00079	HOXB2	1.77432	0.00005
	C4orf47	1.60288	0.00488	NAALADL2-AS2	1.72651	0.00017
	CDKL1	1.56006	0.00018	TEX15	1.66229	0.00932
Downregulation	LMO3	−2.80268	0.00128	ODAM	−2.59037	0.00098
	ABCC6P1	−2.80138	0.00244	ABHD11-AS1	−2.53907	0.00021
	GUCY1B3	−2.80121	0.00211	MAGEC1	−2.36747	0.00118
	TSPEAR-AS1	−2.63897	0.00351	LRRC39	−2.35825	0.00007
	CEMP1	−2.63778	0.00374	NCALD	−2.25073	0.00003
	LOC101929584	−2.48102	0.00399	CD4	−2.23406	0.00861
	CFAP57	−2.02884	0.00452	CECR1	−2.01936	0.00307
	MIR24-1	−1.98172	0.00827	PAX7	−1.95793	0.00975
	LOC102724450	−1.97767	0.00351	HAPLN1	−1.95591	0.00021
	PLA2G10	−1.93472	0.00159	RPLP0P2	−1.84793	0.00017

Genes up- or down-regulated by E2 (10^−9^M) or E4 (10^−7^M) in comparison with vehicle condition obtained by RNAseq analysis, adjusted *p*-value: adj p-value, Fc: Fold change; *n* = 5 replicates per condition.

## Data Availability

RNAseq raw data are available on Gene Expression Omnibus GEO; https://www.ncbi.nlm.nih.gov/geo/query/acc.cgi?acc=GSE173300, GEO accession: GSE173300, public on 17 May 2021.

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
