# Peer review of "Estetrol Combined to Progestogen for Menopause or Contraception Indication Is Neutral on Breast Cancer"

_cancers, 2021, doi:10.3390/cancers13102486_

Round 1

Reviewer 1 Report

In this manuscript, Dr. Gallez and collaborators study the effects of estetrol administration on breast cancer progression in postmenopausal murine models of BC. The authors first show that progesterone and a progestin (DRSP) block the pro-proliferative effects of estradiol (E2) and estetrol (E4) in the endometrium in vivo. Next, they show that estetrol promotes breast cancer in mice in a dose-dependent manner, and that addition of progesterone or DRSP does not significantly impact tumor progression. Then, the authors show that E4 is less potent activating ESR1 than E2, both in vitro and in vivo. Further, they study the transcriptional changes caused by estetrol alone (which are also dose-dependent) and in combination with progesterone or DRSP, which has limited transcriptional impact on their models. Finally, they investigate the differences between E2 and E4 in ESR1 coregulator recruitment.

I enjoyed reading this manuscript. It is well written, clearly presented, contains sound data, and the experiments are properly done and well-controlled. The Material and methods section is comprehensive. It is evident from both the text and the figures that the authors have thought about the experiments carefully and are in control of their work. While some questions remain open and some limitations of the study are pointed out by the authors in the discussion, the final message of this manuscript is solid. Therefore, for the sake of good reviewing practices, I recommend that it is accepted for publication in Cancers with just a few minor comments.

Comments:

1) It is not clear why did the authors choose to show the SEM instead of the SD. For instance, in figure 5, the quantification of the pS118-ERa ovx shows a very low SEM, whereas the western blot image shows some notable differences between the samples, which is visually rather confusing.

2) Please homogenize the font sizes used in the figures and label the x and y axis independently of the graph to achieve an always uniform text (i.e. avoid text distortion due to graph scaling).

3) Please provide higher resolution images for Figure 7.

4) I would suggest to tone down some statements such as "PDX are currently the most powerful tool for understanding breast cancer characteristics and for predicting drug potency". PDXs are typically generated in severely immunocompromised mice; this has many consequences (e.g. lack of major tumor immunity, modified cancer stroma, expansion of cancer stem cells, etc), many of which do not recapitulate aspects of the human disease.

Reviewer 2 Report

This article describes the evaluation of the breast cancer risk associated with a combination of estetrol (E4) with progestogens for the patients with breast cancer and taking progestogens for MHT and COC.  It is known that hormonal treatments increase breast cancer risk, and how to address this drawback is a critical task. In this article, the studies are very well planned: the tumor progression with their signaling pathway was studied. Not only the therapeutic dose of E4 but also overdose was evaluated. The study of lung metastasis is valuable since lung metastasis makes cancer treatment harder.

  • Only one concern is that there are some long sentences. It is known the long sentence decreases the reader’s understanding. For example, there is one sentence in lines 433–436. To improve the readability of the manuscript, I suggest the authors reduce long sentences.
  • The quality of Figure 7 is lower than others. If possible, clear profiles are preferable.
